# Moderating Effects of Telework Intensity on the Relationship Between Ethical Climate, Affective Commitment and Burnout in the Colombian Electricity Sector Amid the COVID-19 Pandemic

**DOI:** 10.3390/bs15101409

**Published:** 2025-10-16

**Authors:** Carlos Santiago-Torner

**Affiliations:** Department of Economics and Business, Faculty of Business and Communication Studies, University of Central University of Catalonia, 08500 Vic, Spain; carlos.santiago@uvic.cat

**Keywords:** telework intensity, burnout, ethical leadership, ethical climate, affective commitment

## Abstract

Background: Ethical leadership and ethical climate are generally considered protective factors against burnout, while affective commitment has traditionally been understood as a personal resource that enhances employee well-being. However, recent evidence suggests that, under specific contextual conditions, these variables may also operate as demands that intensify emotional strain. Objective: This study examines how telework intensity moderates the relationships between ethical leadership, affective commitment, principle-based ethical climate, and burnout. Methods: Data were drawn from a doctoral study conducted in the Colombian electricity sector. Moderation analyses were performed to assess whether the number of telework days per week altered the strength and direction of associations between organizational variables and the dimensions of burnout. Results: Telework intensity did not moderate the relationship between ethical leadership and affective commitment, but it strengthened the positive association between affective commitment and emotional exhaustion. Moreover, it reversed the role of a principle-based ethical climate: from being positively associated with emotional exhaustion and depersonalization to acting as a protective factor under medium to high telework intensity. Conclusions: The findings challenge conventional assumptions about affective commitment and ethical climate, highlighting the ambivalent role of telework. They underscore the need for more nuanced theoretical frameworks and management practices that are sensitive to emerging psychosocial risks in virtual work environments.

## 1. Introduction

The accelerated shift to telework during the COVID-19 pandemic not only transformed daily work routines but also reshaped the psychosocial architecture of organizations ([21]). While working from home is often associated with flexibility, autonomy, and improved work–life balance, research has consistently shown that its consequences for employee well-being are ambivalent ([35]). High telework intensity—commonly defined as the number of days per week an employee works remotely—has been linked to positive outcomes such as reduced commuting stress, but also to adverse effects such as social isolation and emotional exhaustion ([13]; [29]). These contradictions underscore the importance of conceptualizing telework intensity not as inherently beneficial or detrimental, but as a contextual condition that shapes the impact of organizational practices and employee attitudes.

Beyond its operational definition as the number of remote workdays per week, telework intensity can be understood conceptually as the degree of virtualization of the employment relationship. This encompasses the extent to which employees’ daily activities, communication patterns, and boundary management strategies are shaped by remote work arrangements ([1]; [16]; [57]). At higher levels of intensity, employees experience stronger exposure to both resources (e.g., autonomy, time efficiency, reduced commuting) and demands (e.g., digital overload, social isolation, and blurred work–home boundaries). From this perspective, telework intensity functions as a contextual condition that reconfigures how organizational practices and personal resources are enacted and perceived, providing a theoretical rationale for its role as a moderator in this research.

The present study is guided by the Job Demands–Resources (JD-R) model ([4], [5]), which provides a comprehensive framework for understanding how job demands and job resources jointly influence employee well-being and performance. According to the JD-R model, burnout emerges when job demands chronically exceed available resources, whereas resources may buffer strain and foster engagement. Importantly, constructs typically framed as resources—such as affective commitment or ethical climate—can under certain contextual conditions operate as demands, requiring additional energy and thereby intensifying emotional strain ([24]). This theoretical perspective is particularly relevant for telework, where work–home boundary blurring, technology-mediated interactions, and changing expectations can alter how resources and demands are experienced.

Within this evolving landscape, ethical leadership and ethical climate have been identified as key organizational resources that foster trust, psychological safety, and prosocial behavior ([10]; [31]). Ethical leaders promote fairness, transparency, and concern for others, while principle-based ethical climates ([55]) create shared moral expectations that can shield employees from moral conflict and distress ([19]). Similarly, affective commitment—defined as an emotional bond with the organization—has traditionally been regarded as a personal resource that enhances motivation, engagement, and resilience ([8]; [27]; [28]; [47]).

However, recent studies suggest that these resources are not unconditionally protective. Under conditions of sustained telework, affective commitment may paradoxically increase the risk of emotional strain, as highly committed employees devote substantial energy to sustaining relationships and meeting organizational expectations across blurred work–home boundaries ([22]). Likewise, ethical climates, though generally beneficial, may become demanding when employees perceive excessive rigidity without sufficient opportunities for support or dialogue ([7]). In line with the JD-R model, these findings highlight the ambivalent character of constructs that can either buffer or exacerbate strain, depending on contextual factors such as telework intensity.

Despite these insights, little is known about how telework intensity shapes the interplay between ethical leadership, principle-based ethical climate, affective commitment, and burnout. Moreover, prior research has often conceptualized burnout as a unidimensional construct without systematically distinguishing its core components, even though emotional exhaustion and depersonalization are recognized as the most critical dimensions of burnout ([26]; [48]; [50]). This study addresses this gap by focusing specifically on these two dimensions, which are most directly linked to deteriorations in psychological well-being and organizational functioning.

Internationally, research on telework and burnout has proliferated in North America ([1]), Europe ([13]), and Asia ([30]; [57]), yet much less is known about these dynamics in Latin America. In the Colombian context, the electricity sector provides a salient case: it combines responsibility for essential services, 24/7 operational demands, and accelerated digitalization, while simultaneously navigating regulatory and infrastructural challenges that shaped telework implementation during the COVID-19 pandemic ([36]).

By examining whether telework intensity moderates the effects of ethical leadership, affective commitment, and principle-based ethical climate on emotional exhaustion and depersonalization, this study contributes to clarifying whether these constructs operate as stable protective resources or whether, under specific conditions, they may paradoxically contribute to emotional strain. In doing so, it advances knowledge in two ways: first, by extending the JD-R model to account for telework intensity as a boundary condition that can reconfigure the roles of organizational resources and demands, and second, by offering novel evidence from the Colombian electricity sector, where pandemic-induced remote work created a unique setting to analyze these dynamics.

## 2. Theoretical Background and Hypotheses

### 2.1. Ethical Leadership and Affective Commitment

Ethical leadership has been consistently associated with favorable employee attitudes, particularly affective commitment ([2]; [52]). Defined as the emotional attachment and identification employees feel toward their organization ([28]), affective commitment is strengthened when leaders demonstrate integrity, fairness, and concern for others. Through modeling and reinforcing ethical norms, leaders foster trust and fairness perceptions that motivate employees to reciprocate with stronger organizational bonds ([10]; [31]). Meta-analytic evidence further confirms that ethical leadership reliably predicts commitment across diverse organizational contexts ([32]).

From a theoretical standpoint, both social learning and social exchange perspectives explain this relationship. Ethical leaders provide salient behavioral cues that guide employees’ conduct, while simultaneously signaling organizational support and fairness, thereby encouraging reciprocal loyalty ([53]). Within the Job Demands–Resources (JD-R) model, ethical leadership functions as a resource that reduces ambiguity and nurtures socio-emotional support, promoting positive attitudes even in demanding environments ([5]; [24]).

Although research has shown that remote work alters communication patterns and may challenge leader–follower interactions ([57]), recent evidence suggests that ethical leadership remains effective in virtual contexts, as ethical signals can be transmitted through transparent communication and fair decision-making ([15]). However, empirical studies directly examining whether the intensity of telework—measured as the number of days per week employees work remotely—moderates this association remain scarce. This gap underscores the novelty of testing whether the positive effects of ethical leadership on affective commitment are stable across telework intensities.

**H1.** 
*Ethical leadership is positively associated with affective commitment, regardless of the number of teleworking days.*


### 2.2. Affective Commitment and Emotional Exhaustion Under Telework Intensity

Affective commitment has traditionally been framed as a beneficial resource that strengthens employees’ motivation, engagement, and resilience ([28]; [47]). From this perspective, employees who experience strong emotional bonds with their organization are expected to display higher energy and lower vulnerability to burnout. However, recent evidence suggests that affective commitment does not always function as a buffer. Under certain conditions, particularly when work demands intensify, commitment may increase strain by motivating employees to overextend themselves in order to meet organizational expectations ([22]).

The Job Demands–Resources (JD-R) model offers a compelling explanation for this paradox. While commitment may operate as a personal resource, it can also lead employees to invest excessive effort, thereby amplifying the impact of job demands on well-being ([5]). When telework intensity is high, the blurring of work–home boundaries, reduced opportunities for spontaneous interaction, and reliance on digital communication channels require employees to devote additional cognitive and emotional resources to sustain social bonds and respond to organizational norms ([57]). In such contexts, highly committed employees may experience heightened pressure to remain responsive and engaged, inadvertently increasing their vulnerability to emotional exhaustion.

This perspective aligns with findings that commitment-driven overinvestment can foster emotional strain when organizational demands remain high or ambiguous ([45]; [58]). Empirical work on remote and hybrid contexts also highlights that committed employees may face stronger difficulties in detaching from work, as digital environments foster constant connectivity and reinforce expectations of availability ([21]; [29]). These dynamics illustrate that affective commitment, while generally beneficial, may under conditions of intensive telework coexist with or even exacerbate emotional exhaustion.

**H2.** 
*Affective commitment is positively associated with emotional exhaustion, and this positive association strengthens as telework intensity increases.*


### 2.3. Principle-Based Ethical Climate, Emotional Exhaustion, and Depersonalization Under Telework Intensity

Principle-based ethical climates are organizational environments where shared expectations emphasize adherence to moral principles such as fairness, honesty, and responsibility ([55]). These climates provide normative clarity and reinforce a collective sense of justice, which can reduce employees’ exposure to ethical dilemmas and mitigate strain ([31]). Prior research has shown that ethical climates are associated with lower stress and burnout symptoms, as they promote trust, meaning, and moral consistency in the workplace ([7]; [25]).

Within the framework of the Job Demands–Resources (JD-R) model, a principle-based ethical climate functions as an organizational resource that buffers the negative effects of high demands. By fostering shared moral norms, it enhances predictability and psychological safety, which reduce the risk of emotional exhaustion and counteract feelings of depersonalization ([5]; [24]). However, the strength of this protective effect may depend on contextual conditions—particularly telework intensity.

In remote work environments, the absence of informal social cues and the reliance on digital communication can increase the risk of ambiguity and detachment. Under such conditions, the presence of a strong principle-based ethical climate may be particularly valuable, as it provides a stable normative framework that substitutes for diminished face-to-face interactions. Empirical evidence suggests that ethical climates can mitigate burnout by supplying moral consistency and reducing uncertainty ([19]). This effect may be amplified at moderate to high levels of telework intensity, where employees rely more heavily on shared organizational values to navigate reduced social presence.

In contrast, when telework intensity is low, employees can rely more on direct leader interaction and peer support, which may reduce the added value of climate effects for preventing emotional exhaustion. Nevertheless, for depersonalization, the protective influence of principle-based ethical climates is expected to persist across low, moderate, and high telework intensity, since the presence of shared ethical norms continuously reinforces prosocial attitudes and discourages interpersonal withdrawal.

**H3.** 
*A principle-based ethical climate is negatively associated with emotional exhaustion and depersonalization. This negative association is stronger for emotional exhaustion at moderate and high levels of telework intensity, while its protective effect against depersonalization is present across all levels of telework intensity.*


### 2.4. Conceptual Model: Ethical Leadership, Ethical Climate of Principles, Affective Commitment and Burnout: Moderated by Telework Intensity

The model (Figure 1) illustrates the proposed associations between ethical leadership, affective commitment, principle-based ethical climate, and the two core dimensions of burnout: emotional exhaustion and depersonalization. Hypothesis 1 (H1) predicts that ethical leadership is positively associated with affective commitment, independently of telework intensity. Hypothesis 2 (H2) posits that affective commitment is positively related to emotional exhaustion, with this association becoming stronger as telework intensity increases. Hypothesis 3 (H3) proposes that a principle-based ethical climate is negatively related to emotional exhaustion and depersonalization. For emotional exhaustion, the protective effect of ethical climate is expected to be stronger at moderate and high levels of telework intensity, while its protective effect against depersonalization is expected to hold across all levels of telework intensity. Telework intensity thus operates as a moderator of the associations between affective commitment and emotional exhaustion, as well as between ethical climate and burnout.

## 3. Materials and Methods

### 3.1. Participants and Sampling Strategy

The research was conducted within the Colombian electricity sector, a context that concentrates a highly qualified workforce due to the technical and professional requirements of most positions. The population of interest comprised employees whose roles involved creativity and problem-solving as central tasks, while operational staff with limited cognitive demands were excluded ([39], [40]).

A multi-stage probability sampling design was adopted to maximize representativeness. First, five major urban centers—Bogotá, Medellín, Cali, Pereira, and Manizales—were selected as geographical clusters given their relevance for the sector. Within these clusters, six organizations agreed to participate: EPM, ISA, CIDET, XM, CHEC, and DISPAC. Employees from these organizations were then randomly invited to participate. The minimum required sample size was estimated at 382 participants (95% confidence level), and a total of 448 valid responses were obtained, thus ensuring sufficient statistical power for the analyses.

The sample included 273 men (61%) and 175 women (39%). Most participants were under 50 years old (82%), had postgraduate qualifications (60%), and held permanent contracts (81%). The distribution by position was: analysts (69%), support staff (17%), middle managers (9%), and directors (5%). Average telework intensity was 3.3 days per week, with approximately 28% reporting full-time remote work.

### 3.2. Data Collection Procedure

Data were collected between November and December 2021 through an online questionnaire administered via Microsoft Forms. The study was introduced to organizations through the collective action platform of the sector, and participation was voluntary. Prior to completing the survey, participants were provided with information regarding the purpose of the research, confidentiality guarantees, and withdrawal rights. Data collection sessions were carried out during working hours, with the presence of the researcher to clarify potential questions.

### 3.3. Measures

Validated instruments widely used in international research were adapted to the Colombian context through a translation and back-translation procedure ([9]). Two bilingual experts independently translated the original English scales into Spanish, and two additional experts, blind to the originals, back-translated them. Discrepancies were discussed until full semantic and conceptual equivalence was achieved. A pilot test with 25 employees from the electricity sector confirmed clarity and contextual adequacy. Construct validity was assessed through confirmatory factor analyses (CFA), with all standardized factor loadings above 0.60 (except for one item from the depersonalization subscale, which was removed). Internal consistency was assessed with Cronbach’s alpha coefficients, which exceeded the recommended threshold of 0.70 for all constructs.

Ethical Leadership (Independent Variable). Measured with the 10-item unidimensional Ethical Leadership Scale developed by [11] ([11]). Sample item: “My supervisor disciplines employees who violate ethical standards.” Cronbach’s α = 0.92.

Principle-Based Ethical Climate (Independent Variable). Assessed with the 11-item measure by [55] ([55]), capturing the extent to which decision-making is guided by moral principles, rules, and professional codes. Example item: “In this organization, the first consideration is whether a decision violates any law.” Cronbach’s α = 0.74.

Affective Commitment (Dependent, Independent, and Mediating Variable). Measured with the 8-item affective commitment subscale by [28] ([28]). Example item: “I feel a strong sense of belonging to my organization.” Cronbach’s α = 0.86.

Burnout (Dependent Variables). Evaluated using the Maslach Burnout Inventory–General Survey (MBI–GS; [46]). Emotional exhaustion was measured with five items (α = 0.90), e.g., “I feel emotionally drained from my work.” Depersonalization was assessed with five items (α = 0.90). One item from the depersonalization subscale (item 13) was removed due to low factor loading (<0.40).

Telework Intensity (Moderator). Operationalized as the self-reported number of telework days per week (ranging from 1 to 5), consistent with prior studies ([54]; [56]). This operationalization captures the degree to which remote work structures employees’ weekly schedules and provides a continuous measure of telework exposure.

All items (except telework intensity) were rated on a 6-point Likert-type scale ranging from 1 (“strongly disagree”) to 6 (“strongly agree”).

### 3.4. Ethical Considerations

The research protocol was reviewed and approved by the Ethics Committee of the University of Vic—Central University of Catalonia (20 July 2021; code 170/2021). The evaluation confirmed compliance with ethical guidelines, including informed consent and data confidentiality, in line with the Code of Good Scientific Practice of the Spanish Ministry of Science and Innovation.

### 3.5. Analytical Strategy

The hypotheses were tested using moderation analyses conducted with Hayes’s PROCESS macro for SPSS version 27 (Model 1; [17]). In these models, ethical leadership and principle-based ethical climate were specified as independent variables, affective commitment as both a dependent variable (H1) and an independent predictor (H2), and emotional exhaustion and depersonalization as dependent variables (H2, H3). Telework intensity (measured as the number of teleworking days per week) was entered as a continuous moderator of the focal associations. Gender, age, and organizational tenure were included as covariates. Each moderation was tested through hierarchical regression models including the main effects and the interaction term (independent variable × telework intensity). Continuous predictors were mean centered before computing the interaction terms. For all models, 10,000 bootstrap samples were generated to derive bias-corrected confidence intervals for the interaction effects, thereby increasing robustness against violations of normality. Statistical significance was set at *p* < 0.05.

To ensure measurement quality, the psychometric properties of all constructs—including reliability, convergent and discriminant validity, and inter-variable correlations—were rigorously assessed as part of the doctoral dissertation project from which this article derives ([41]). Confirmatory factor analysis (CFA) supported the distinctiveness of the six focal variables, with acceptable model fit indices (χ^2^/df < 3, CFI > 0.90, TLI > 0.90, RMSEA < 0.08). Since these results have already been documented elsewhere, the present manuscript focuses exclusively on the moderation analyses and their implications, which have not been previously disseminated.

## 4. Results

### 4.1. Ethical Leadership and Affective Commitment

The moderation analysis indicated that telework intensity did not significantly alter the relationship between ethical leadership and affective commitment. Ethical leadership showed a positive and significant association with affective commitment (b = 0.25, *p* = 0.001), while the interaction term with telework intensity was non-significant (b = 0.02, *p* = 0.670). These results confirm H1, supporting the hypothesis that ethical leadership is positively associated with affective commitment, independently of the number of teleworking days. (see Table 1).

### 4.2. Affective Commitment and Emotional Exhaustion

A different pattern was observed when examining the relationship between affective commitment and emotional exhaustion. The interaction between affective commitment and telework intensity was significant (b = 0.27, *p* < 0.05). Simple slope analyses showed that the positive association between affective commitment and emotional exhaustion became stronger as the number of telework days increased. As illustrated in Table 2 and Figure 2, these findings suggest that in intensive telework contexts, affective commitment does not function as a protective resource against burnout but may instead coexist with, or even amplify, emotional exhaustion, given the additional effort required to sustain social interactions and respond to heightened ethical demands. These results provide empirical support for H2.

### 4.3. Ethical Climate and Emotional Exhaustion

Telework intensity significantly moderated the association between a principle-based ethical climate and emotional exhaustion. At low levels of telework intensity, a principle-based ethical climate was positively associated with emotional exhaustion (b = 0.20, *p* < 0.01). However, as telework intensity increased, this relationship reversed and became negative, indicating that a strong ethical climate acted as a protective factor against emotional exhaustion under more intensive telework conditions (see Table 3 and Figure 3). These findings provide partial support for H3, suggesting that the protective role of ethical climate becomes more salient at moderate and high levels of telework.

### 4.4. Ethical Climate and Depersonalization

A consistent negative association was observed between principle-based ethical climate and depersonalization across low, medium, and high levels of telework intensity. As shown in Table 4 and Figure 4, the interaction term was significant (b = −0.05, *p* < 0.05), but the direction of the effect indicates that the protective function of ethical climate remains robust regardless of telework intensity. These findings confirm H3, highlighting that ethical climates emphasizing principles can reduce depersonalization even when employees face varying degrees of remote work.

In conclusion, telework intensity emerged as a key contextual factor that reshaped the examined relationships. While ethical leadership consistently fostered affective commitment across all work modalities, high telework intensity amplified the emotional costs of commitment. At the same time, principle-based ethical climates shifted from neutral or even adverse conditions under low telework intensity to protective resources against burnout at moderate to high levels.

## 5. Discussion

The present study advances understanding of how telework intensity moderates the interplay between ethical leadership, affective commitment, principle-based ethical climate, and burnout. Grounded in the Job Demands–Resources (JD-R) model ([5]), the findings highlight that telework functions as a contextual condition that can amplify or mitigate the effects of organizational resources, rather than as a uniform demand or benefit. This theoretical lens enables a more precise interpretation of the results, underscoring the ambivalence of telework as both a potential enabler and stressor in shaping employee well-being.

First, the results confirmed that telework intensity does not moderate the relationship between ethical leadership and affective commitment, reinforcing the robustness of this link across work modalities. Ethical leadership behaviors—characterized by fairness, reliability, and principled decision-making ([2]; [10]; [33])—strengthen employees’ affective bonds with the organization through mechanisms of social exchange ([14]; [20]; [37], [38]). From a JD-R perspective, ethical leadership consistently functions as a resource that fosters trust, reciprocity, and psychological safety, thereby sustaining affective commitment irrespective of telework intensity. This finding contributes to the literature by confirming the stability of the ethical leadership–commitment relationship in both face-to-face and virtual contexts, a dimension that prior research has rarely examined.

Second, a counterintuitive pattern emerged regarding affective commitment and emotional exhaustion. Consistent with H2, affective commitment was positively associated with exhaustion, with the effect intensifying under conditions of high telework intensity. Traditionally framed as a resource that energizes employees ([28]; [47]), affective commitment may under intensive telework become a demand. Highly committed employees mobilize considerable effort to sustain social ties and comply with ethical expectations, yet technological mediation often fails to fully satisfy needs for relatedness ([22]; [23]). This “resource–demand shift” aligns with JD-R assumptions, demonstrating that resources can transform into stressors when excessive investment is required to maintain them ([6]). Theoretically, this challenges the conventional dichotomy between commitment and burnout, showing that both may coexist simultaneously in teleworking contexts.

Third, the study provides a nuanced understanding of principle-based ethical climates. As expected in H3, telework intensity moderated their relationship with emotional exhaustion. At low levels of telework, an ethical climate was weakly or even positively associated with exhaustion, suggesting that overly rigid moral structures may create unintended pressures ([7]; [42], [44]; [12]). However, at moderate and high telework intensity, this association reversed, with ethical climate acting as a protective factor. Reduced physical exposure to ethical tensions and clearer virtual role structures may buffer the strain otherwise produced by normative rigidity ([18]; [34]). In JD-R terms, the climate functions as a contextual resource whose protective effects are activated under specific work conditions.

Finally, the consistent negative association between principle-based ethical climate and depersonalization across all telework levels further underscores its resource role. Shared ethical values provide a moral anchor that sustains empathy and prevents emotional distancing, even in technology-mediated environments ([49]; [51]). This aligns with prior findings that highlight the cohesion-building function of organizational ethics, which foster belonging and identification while reducing cynicism ([3]; [19]; [43]).

Taken together, the findings contribute to the literature in three ways. First, they empirically validate the JD-R proposition that resources and demands are not static categories but may shift depending on contextual moderators such as telework intensity. Second, they advance theoretical understanding by clarifying the ambivalent role of affective commitment, showing it can simultaneously energize and exhaust employees under intensive telework. Third, they extend research on ethical climates by revealing that their protective potential becomes more salient when employees operate in hybrid or virtual contexts.

From a practical perspective, these results suggest that organizations should not assume that traditional resources such as commitment or ethics are unconditionally beneficial. Instead, managers in sectors with high telework exposure—such as the electricity industry—should adapt human resource strategies to balance demands and resources dynamically. Ethical leadership training, flexible ethical standards that encourage dialogue rather than rigidity, and support mechanisms to prevent the overextension of highly committed employees may prove critical to mitigating burnout risks.

### 5.1. Theoretical Implications

The findings of this study contribute to advancing theoretical debates on ethical leadership, ethical climate, and telework by emphasizing the complex and ambivalent role that work context plays in shaping organizational outcomes.

First, the confirmation that telework intensity does not moderate the positive relationship between ethical leadership and affective commitment reinforces the context-independent validity of social exchange theory. Ethical leadership behaviors—grounded in integrity, fairness, and reciprocity—appear to foster affective commitment regardless of whether interactions occur in physical or virtual settings. This suggests that the relational mechanisms proposed by social exchange theory remain robust in digitally mediated environments, extending its applicability to hybrid and remote work contexts.

Second, the counterintuitive finding that affective commitment is positively associated with emotional exhaustion, particularly under high telework intensity, challenges the traditional conceptual dichotomy between commitment and burnout. These results imply that affective commitment can simultaneously function as a resource and as a demand, depending on contextual conditions. This insight aligns with, but also extends, the Job Demands–Resources framework by showing that under telework-intensive conditions, affective commitment may shift from being a protective resource to becoming a strain-inducing demand. This dual role calls for a reconceptualization of affective commitment in remote work settings, moving away from linear assumptions toward a more dynamic and context-contingent perspective.

Third, the evidence that a principle-based ethical climate can act either as a source of strain or as a protective factor highlights the conditional nature of ethical climates. While rigid moral environments may heighten moral distress and psychological strain in traditional settings, high telework intensity appears to mitigate these risks and transform principle-based climates into resources that protect against burnout. This finding suggests the need to refine ethical climate theory by incorporating contextual moderators—such as work modality—that shape whether ethical norms are experienced as supportive or constraining.

Finally, the robust negative association between principle-based ethical climates and depersonalization across all levels of telework intensity underscores the cohesive and integrative function of shared ethical values in virtual work contexts. This supports the argument that ethical climates can serve as anchors of organizational identification and moral orientation, even when opportunities for social interaction are reduced. Thus, the study adds to emerging theories of virtual work by showing how shared values, rather than physical proximity, may constitute the primary drivers of cohesion and empathy in digital environments.

In sum, the study contributes to theory by (a) reaffirming the contextual resilience of social exchange theory, (b) problematizing traditional assumptions regarding affective commitment as purely protective, (c) extending ethical climate theory toward a conditional and context-sensitive framework, and (d) integrating telework intensity as a pivotal boundary condition in models of organizational behavior.

### 5.2. Practical Implications

The results of this study provide actionable insights for managers and human resource practitioners in the Colombian electricity sector and other industries characterized by high responsibility, continuous operations, and increasing reliance on hybrid and remote work.

First, the consistent positive association between ethical leadership and affective commitment, regardless of telework intensity, underscores the importance of institutionalizing ethical leadership as a core managerial competency. For electricity companies where trust and safety are critical, leadership training programs should go beyond compliance with codes of conduct and emphasize relational skills such as transparency, compassion, and consistency. Embedding these skills into leadership pipelines and performance evaluation systems can help sustain commitment in both in-person and virtual contexts, where informal trust-building opportunities are limited.

Second, the finding that affective commitment can simultaneously enhance emotional exhaustion under high telework intensity reveals a potential paradox for employee well-being. In high-demand industries, managers must carefully monitor digital workload and communication intensity. Practical measures include setting explicit limits on after-hours digital availability, encouraging “digital recovery” practices such as scheduled offline periods, and introducing workload audits to identify employees at risk of over-engagement. For example, electricity companies could implement real-time dashboards to monitor overtime patterns or use pulse surveys to detect early signs of exhaustion among highly committed employees.

Third, the ambivalent role of principle-based ethical climates suggests the need for flexible, context-sensitive application of ethical norms. Rather than enforcing rigid codes that may inadvertently increase stress, managers should frame ethical principles as enabling guidelines. In practice, this may involve incorporating short ethical decision-making scenarios into virtual safety briefings, creating digital consultation spaces for employees to discuss moral dilemmas, and embedding supportive—not punitive—ethical reminders in collaboration platforms. These strategies ensure that ethics remain a resource for guidance rather than a source of pressure.

Fourth, the consistent protective effect of principle-based ethical climates on depersonalization across telework intensities highlights the importance of leveraging shared values to maintain empathy and organizational identification. In sectors where social cohesion directly affects service continuity, organizations could introduce rituals such as beginning virtual meetings with brief reflections on core values, recognizing employees for integrity-driven actions, and aligning reward systems with ethical behavior. Such practices reinforce moral identity and counteract relational erosion often observed in remote teams.

Finally, the broader implication is that telework intensity must be explicitly treated as a boundary condition in HR design. Rather than adopting “one-size-fits-all” strategies, organizations should recalibrate leadership practices, ethical frameworks, and well-being initiatives depending on the intensity of telework in each unit. For example, teams with higher telework exposure may require stronger virtual ethics training and digital recovery protocols, while those working primarily on-site may benefit from more traditional safety and trust-building practices.

In summary, organizational leaders should: (a) institutionalize ethical leadership as a relational competency; (b) monitor the double-edged nature of affective commitment; (c) frame ethical climates as supportive rather than prescriptive; (d) reinforce shared values to sustain empathy in virtual contexts; and (e) incorporate telework intensity as a design parameter in HR and leadership policies. These targeted interventions provide a roadmap for building healthier, more resilient organizations in the electricity sector and beyond in an era of intensified digital work.

### 5.3. Limitations and Future Research Directions

This study, while contributing novel insights into the ambivalent role of telework intensity in the relationships between ethical leadership, principle-based ethical climates, and burnout, is not without limitations. Acknowledging these constraints is essential to contextualize the findings and guide future inquiry.

First, the reliance on a cross-sectional design restricts the ability to draw strong causal inferences. Although the theoretical frameworks employed—social exchange theory and the Job Demands–Resources (JD-R) model—offer plausible explanations for the observed patterns, longitudinal or experimental designs would allow researchers to capture how the dynamics between ethical leadership, telework intensity, and burnout evolve over time.

Second, the data were collected in a single national context—the Colombian electricity sector—which may limit the generalizability of the results. Telework intensity and perceptions of ethical leadership are likely influenced by cultural norms, regulatory frameworks, and organizational traditions. Cross-cultural or multinational studies would be valuable to examine potential differences between collectivist and individualist contexts, as well as across industries with varying degrees of telework adoption.

Third, while this study concentrated on two critical dimensions of burnout—emotional exhaustion and depersonalization—it did not address the full spectrum of employee well-being. Future research should include variables such as work engagement, psychological detachment, or technostress, which could enrich understanding of how telework intensity simultaneously generates resources and demands.

Fourth, the exclusive reliance on self-reported measures raises the possibility of common method bias. Although methodological safeguards were implemented, future studies should incorporate multisource or multimethod designs, such as supervisor ratings, peer assessments, or digital trace data, to enhance the robustness of the findings.

Fifth, telework intensity was operationalized as a quantitative indicator (i.e., number of telework days per week), without fully capturing qualitative dimensions of remote work such as autonomy, task interdependence, or digital overload. Future studies could explore whether particular configurations of remote work conditions moderate the effects of ethical leadership and ethical climates in more nuanced ways.

Finally, the unexpected positive association between affective commitment and emotional exhaustion under telework conditions calls for deeper theoretical exploration. Future research should investigate the mechanisms behind this paradox, including boundary blurring, emotional regulation strategies, and virtual presenteeism. Such inquiries would help clarify whether affective commitment, under certain conditions, transforms from a motivational resource into an emotional demand.

In sum, advancing this line of research requires longitudinal, cross-cultural, multimethod, and multidimensional approaches. By addressing these gaps, future studies can refine our understanding of how ethical leadership and ethical climates interact with telework intensity, thereby contributing to a more comprehensive and context-sensitive framework for managing psychological well-being in hybrid and digital workplaces.

## 6. Conclusions

This study advances knowledge on the complex interplay between ethical leadership, principle-based ethical climates, affective commitment, and burnout, highlighting the ambivalent role of telework intensity as a contextual factor.

The findings demonstrate that ethical leadership consistently fosters affective commitment, regardless of telework intensity, underscoring the robustness of ethical conduct as a driver of trust and organizational attachment. At the same time, the study reveals a paradoxical outcome: affective commitment, typically regarded as a protective factor, is positively associated with emotional exhaustion, with the relationship becoming particularly pronounced under conditions of intensive telework. This counterintuitive finding shows that commitment can, under certain circumstances, function as a double-edged sword, intensifying emotional strain when relational needs remain unmet and ethical demands become more salient.

The results further demonstrate that telework intensity moderates the role of principle-based ethical climates in significant ways. While rigid ethical environments may foster stress under low telework intensity, higher levels of telework appear to transform such climates into protective resources, mitigating emotional exhaustion and preventing moral overstrain. Moreover, principle-based ethical climates consistently reduced depersonalization across all telework levels, highlighting their role in anchoring empathy and sustaining identification with the organization even in virtual contexts.

Taken together, these findings challenge simplified views of telework as either a resource or a demand. Instead, they illustrate its ambivalent nature, showing how telework intensity reconfigures the way established organizational factors—ethical leadership, affective commitment, and ethical climates—translate into well-being or strain. Recognizing this complexity is crucial for both scholars and practitioners aiming to understand and manage the psychological consequences of work in increasingly hybrid and digital environments.

## Figures and Tables

**Figure 1 behavsci-15-01409-f001:**
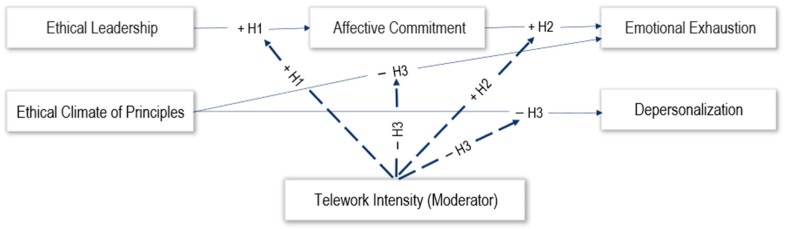
Conceptual model of the hypothesized relationships.

**Figure 2 behavsci-15-01409-f002:**
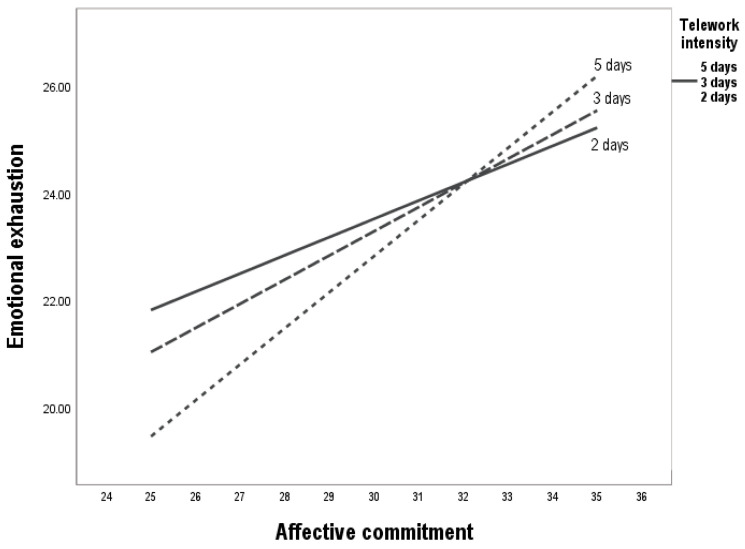
Effect of telework intensity on the relationship between affective commitment and emotional exhaustion. Note: The figure illustrates that affective commitment consistently predicts higher levels of emotional exhaustion, and this association is progressively amplified as telework intensity increases (low, medium, high). This finding highlights the paradoxical role of affective commitment, which, under conditions of greater virtual work, may transform from a resource into a psychological demand that exacerbates strain.

**Figure 3 behavsci-15-01409-f003:**
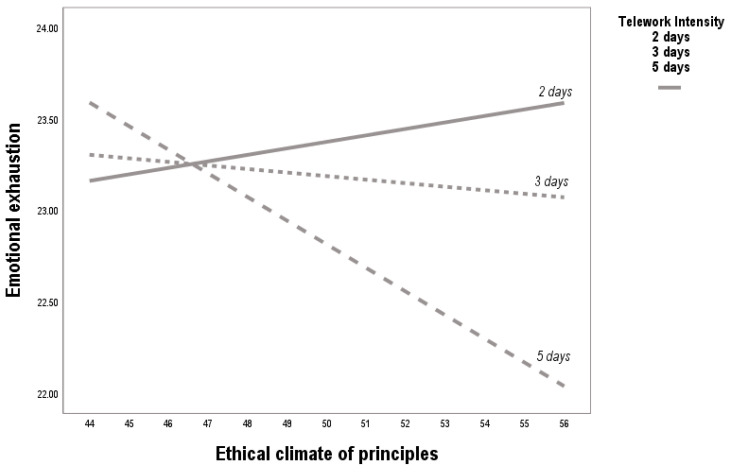
Effect of telework intensity on the relationship between ethical climate of principles and emotional exhaustion. Note: At low telework intensity, a principled ethical climate is positively associated with emotional exhaustion, suggesting that excessive normative expectations can be psychologically taxing. However, under medium and high telework intensity, this relationship reverses, with the ethical climate acting as a protective factor that buffers emotional exhaustion. This moderation underscores the ambivalent role of telework in reshaping the effects of ethical contexts.

**Figure 4 behavsci-15-01409-f004:**
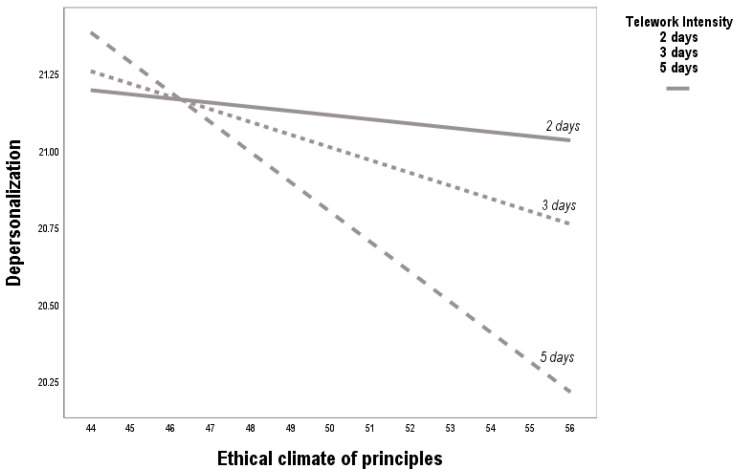
Effect of telework intensity on the relationship between ethical climate of principles and depersonalization. Note: The figure shows that telework intensity consistently reverses the positive relationship between principled ethical climate and depersonalization. Across low, medium, and high levels of telework, the ethical climate functions as a buffer, mitigating emotional distancing and cynicism toward colleagues and clients. This suggests that in virtualized work environments, shared moral principles can foster cohesion and reduce the risk of depersonalization.

**Table 1 behavsci-15-01409-t001:** Moderation of Telework Intensity Between Ethical Leadership and Affective Commitment.

Model Summary	R	R^2^	ΔR^2^	F (df1, df2)	*p*
Outcome variable:	Affective commitment (Y)				
Overall model	0.29	0.09	0.02	14.01 (3, 443)	<0.01
**Predictor**	**b**	**SE**	**t**	** *p* **	**95% CI [LLCI, ULCI]**
Constant	22.50	3.17	7.01	0.01	[16.27, 28.74]
Ethical Leadership (X)	0.25	0.06	3.42	0.01	[0.11, 0.39]
Telework Intensity (W)	−0.59	1.01	−0.56	0.57	[−1.60, 3.21]
Interaction (X × W)	0.02	0.02	0.43	0.67	[−0.07, 0.05]

Note. Affective commitment was regressed on ethical leadership (X), telework intensity (W), and their interaction term (X × W). Ethical leadership was a significant positive predictor of affective commitment (b = 0.25, *p* = 0.001). The interaction did not reach statistical significance (b = 0.02, t = 0.43, *p* = 0.67). N = 448.

**Table 2 behavsci-15-01409-t002:** Moderation of Telework Intensity Between Affective Commitment and Emotional Exhaustion.

Model Summary	R	R^2^	ΔR^2^	F (df1, df2)	*p*
Outcome variable:	Emotional Exhaustion (Y)				
Overall model	0.46	0.21	0.08	39.09 (3, 444)	<0.01
**Predictor**	**b**	**SE**	**t**	** *p* **	**95% CI [LLCI, ULCI]**
Constant	15.46	3.65	4.23	0.01	[8.29, 22.64]
Affective Commitment (X)	0.27	0.12	2.25	0.02	[0.04, 0.51]
Telework Intensity (W)	−2.49	1.08	−2.29	0.02	[−4.62, −0.35]
Interaction (X × W)	0.08	0.04	2.14	0.03	[0.06, 0.15]

Note. Emotional exhaustion was regressed on affective commitment (X), telework intensity (W), and their interaction term (X × W). The interaction is statistically significant, b = 0.08, t = 2.14, *p* = 0.03. (N = 448).

**Table 3 behavsci-15-01409-t003:** Moderation of Telework Intensity Between Climate of Principles and Emotional Exhaustion.

Model Summary	R	R^2^	ΔR^2^	F (df1, df2)	*p*
Outcome variable:	Emotional Exhaustion (Y)				
Overall model	0.46	0.22	0.09	24.58 (5, 442)	<0.01
**Predictor**	**b**	**SE**	**t**	** *p* **	**95% CI [LLCI, ULCI]**
Constant	24.53	4.30	4.13	0.01	[4.32, 23.69]
Climate of principles (X)	0.20	0.07	3.42	0.01	[0.11, 0.39]
Telework Intensity (W)	−2.72	1.46	−3.66	0.02	[−1.60, −0.21]
Interaction (X × W)	−0.06	0.03	−3.95	0.02	[−0.11, −0.02]

Note. Emotional exhaustion was regressed on climate of principles (X), telework intensity (W), and their interaction term (X × W). The interaction is statistically significant, b = −0.06, t = −3.95, *p* = 0.02. (N = 448).

**Table 4 behavsci-15-01409-t004:** Moderation of Telework Intensity Between Climate of Principles and Depersonalization.

Model Summary	R	R^2^	ΔR^2^	F (df1, df2)	*p*
Outcome variable:	Depersonalization (Y)				
Overall model	0.58	0.34	0.12	35.79 (5, 442)	<0.01
**Predictor**	**b**	**SE**	**t**	** *p* **	**95% CI [LLCI, ULCI]**
Constant	24.41	4.17	4.28	0.01	[4.10, 23.01]
Climate of principles (X)	0.26	0.07	4.32	0.01	[0.21, 0.79]
Telework Intensity (W)	−2.12	1.66	−3.26	0.02	[−1.16, −0.31]
Interaction (X × W)	−0.05	0.02	−3.25	0.02	[−0.42, −0.12]

Note. Depersonalization was regressed on climate of principles (X), telework intensity (W), and their interaction term (X × W). The interaction is statistically significant, b = −0.05, t = −3.25, *p* = 0.02. (N = 448).

## Data Availability

The original data presented in the study and the questionnaire used are openly available at The Open Science Framework repository at https://osf.io/w2g5b/?view_only=f8b9995262ed469eab5413f302dd83c4, accessed on 23 January 2025. In addition, the doctoral thesis can be consulted via the following link: https://www.tdx.cat/handle/10803/694925?locale-attribute=es, accessed on 23 January 2025.

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
