# Peer review of "Moderating Effects of Telework Intensity on the Relationship Between Ethical Climate, Affective Commitment and Burnout in the Colombian Electricity Sector Amid the COVID-19 Pandemic"

_behavsci, 2025, doi:10.3390/bs15101409_

Round 1
Reviewer 1 Report
Comments and Suggestions for Authors
The manuscript investigates the moderating role of telework intensity on the relationships between ethical leadership, affective commitment, principle-based ethical climate, and burnout. By examining whether the number of telework days per week alters the direction and strength of these associations, the study addresses a timely and relevant issue. The findings reveal that the positive relationship between affective commitment and emotional exhaustion intensifies with telework intensity, while the link between ethical leadership and affective commitment remains stable. Moreover, a principled ethical environment appears to act as a protective factor against emotional exhaustion and depersonalization at moderate to high levels of telework, inverting its usual effect.
The manuscript demonstrates a number of strengths. It addresses a highly relevant and underexplored issue: how telework conditions reshape organizational dynamics and affect employee burnout. The rationale for focusing on the electricity sector is clearly presented, and the discussion thoughtfully considers both practical and theoretical implications, particularly in relation to strategies to prevent burnout in teleworking contexts.
Areas for Improvement
- Theoretical Framework: The manuscript lacks a clearly articulated theoretical foundation. It remains unclear whether the central focus is on burnout or teleworking. The hypotheses are not sufficiently grounded in established theory, and the rationale for selecting dependent, independent, and moderating variables needs further elaboration. For instance, the discussion introduces the Job Demands–Resources (JD-R) framework, but the paper would benefit from integrating this (or another framework) at the outset to justify the hypotheses and research model.
- Hypotheses and Research Model: The study would be strengthened by explicitly stating the hypotheses and presenting the conceptual model earlier in the paper (ideally at the end of the introduction or within a theoretical section). This would help clarify the research aims and prevent overlap between main and secondary objectives.
- Measurement and Validity: The manuscript does not provide sufficient detail regarding the measurement instruments. Were the scales translated? If so, what translation principles were applied (e.g., back-translation)? Was construct validity tested? Including results of factor analyses and factor loadings would enhance the methodological robustness and transparency.
- Results Presentation: The results section is somewhat difficult to follow. Reorganizing the findings to align more directly with the research aims and hypotheses would improve clarity. At present, the primary and secondary objectives appear to be conflated. A structured presentation (hypothesis by hypothesis) would strengthen the narrative.
- Discussion: Although the discussion provides broad reflections, it would benefit from closer alignment with the stated aims and hypotheses. For example, references to JD-R theory appear only at this stage, without having guided variable selection earlier. Integrating theoretical considerations throughout the paper would provide stronger coherence and academic contribution.
This is an interesting and relevant study that addresses the intersection of teleworking, organizational climate, and employee burnout. While the manuscript is well-written and methodologically transparent, it would benefit from a stronger theoretical foundation, clearer articulation of hypotheses, and enhanced clarity in the presentation of results. With revisions along these lines, the paper has the potential to make a significant and meaningful contribution to the literature on telework and could offer valuable insights for both researchers and practitioners exploring employee burnout.
Author Response
"Please see the attachment."

Reviewer 2 Report
Comments and Suggestions for Authors
Thank you for the opportunity to review this manuscript. I offer the following feedback to strengthen the research:
- Given that this study was conducted in the Colombian electricity sector during the COVID-19 pandemic, these important contextual elements should be incorporated into the title to better reflect the research scope.
- Introduction: The statement that 'prior research has often conceptualized burnout globally, without distinguishing its core dimensions' (p. 2, line 65) needs to be supported with appropriate citations from the burnout literature to substantiate this methodological critique. The assertion regarding contributions made 'in North American, European, and increasingly Asian contexts' (p. 2, line 80) requires supporting citations that document telework-burnout research across these geographical areas to substantiate this claim. As this study specifically investigates telework burnout in the Colombian electricity sector, the background section should include detailed contextual information about the working conditions under COVID-19 pandemic, telework implementation patterns, and sector-specific factors that may influence burnout experiences in this particular field. Following a comprehensive review of related research, please formulate specific research questions or hypotheses that clearly articulate the expected moderation effects and provide theoretical justification for these relationships.
- Measures and Analytical strategy: The study includes six variables. The manuscript should clearly specify the role and characteristics of each variable, indicating which function as independent variables, dependent variables, and moderating variables. Additionally, the manuscript should clearly describe the specific moderation analyses planned for the study variables, including telework intensity, ethical leadership, affective commitment, principle-based ethical climate, and burnout dimensions.
Author Response
"Please see the attachment."

Reviewer 3 Report
Comments and Suggestions for Authors
Thank you for the opportunity to review this engaging manuscript. The study tackles an important question and offers promising theoretical and practical insights. My comments below are offered aimed at further improving the paper’s contribution and enhancing its clarity for readers.
- Although the manuscript is based on a doctoral dissertation and offers intriguing data, the theoretical contribution needs to be more clearly stated. It is unclear at this point how this work adds to the body of knowledge already established by earlier research on telework, affective commitment, and ethical climate. By doing this, the paper's originality would be reinforced and its contribution would not be viewed as a mere synopsis of dissertation findings.
- There is not much in the theoretical development area. Although pertinent literature is reviewed in the opening, hypotheses are not stated properly. Explicitly presenting research hypotheses (e.g., H1, H2, etc.) based on theoretical reasoning would boost the text for a study using moderation analyses. Instead of coming across as an experimental report, this would make it clear how the work expands upon and grows upon earlier models.
- Although the manuscript is well structured, the inclusion of a conceptual model would strengthen it. A figure summarizing the hypothesized relationships would help readers quickly grasp the study’s framework and appreciate its theoretical contribution.
- Some interpretations risk overstating causality (“telework intensity transforms climate into a protective factor”). This should be rephrased more cautiously.
- Practical implications are broad but sometimes generic (e.g., “develop adaptive HR frameworks”). More concrete, context-specific guidance would be valuable.
- Occasional awkward phrasing (e.g., “revise Table 1” appears in text (line 164), suggesting editing issues).
- Language editing to remove redundancies and improve clarity is also necessary.
Author Response
"Please see the attachment."

Round 2
Reviewer 1 Report
Comments and Suggestions for Authors
The author has undertaken a thorough and meticulous revision of the manuscript, effectively addressing the comments and suggestions raised in the previous review. The revisions substantially enhance the clarity, coherence, and scholarly contribution of the study, positioning it well for publication.
Major Revisions and Enhancements:
- Introduction: The study’s purpose has been clearly redefined, presenting a compelling rationale and situating the research firmly within the extant literature. The introduction now provides a sharper focus and a more convincing justification for the investigation.
- Theoretical Background and Hypotheses: The theoretical framework has been significantly expanded, integrating relevant literature with greater depth and sophistication. Hypotheses are now logically structured, reflecting a cohesive and well-articulated conceptual model.
- Measures: The manuscript now offers comprehensive descriptions of the measurement instruments and procedures, improving transparency and allowing readers to evaluate the validity and reliability of the study more effectively.
- Discussion and Practical Implications: The discussion section has been thoughtfully restructured. Findings are interpreted in a nuanced manner, clearly linked to the theoretical framework, and practical implications are articulated in a way that enhances their relevance to both research and practice.
Minor but Recommended Adjustments:
- Title: For clarity and precision, it is recommended to revise the title from “Moderating Effects of Telework Intensity on Ethical Climate, Affective Commitment and Burnout…” to “Moderating Effects of Telework Intensity on the Relationship between Ethical Climate, Affective Commitment, and Burnout…”. This revision better reflects the nature of the study and aligns with conventional academic phrasing.
- Synthesis of Findings Section: The current section titled “Synthesis of Findings” appears redundant. Its content could be integrated into a concluding paragraph prefaced with “In conclusion”, thereby enhancing cohesion and readability.
Overall Assessment:
The manuscript has undergone substantial and thoughtful improvement, demonstrating careful consideration of prior reviewer feedback. The revisions enhance both the theoretical contribution and practical significance of the study. Subject to the minor adjustments suggested above, the manuscript is now suitable for publication.
Author Response
"Please see the attachment."

Reviewer 2 Report
Comments and Suggestions for Authors
All my concerns from the previous review have been addressed. The author has made appropriate revisions that have improved the manuscript's quality.
Author Response
"Please see the attachment."
